# Methane in Gas Shows from Boreholes in Epigenetic Permafrost of Siberian Arctic

**Gleb Kraev** [1,2,*], **Elizaveta Rivkina** [1], **Tatiana Vishnivetskaya** [1,3], **Andrei Belonosov** [4], **Jacobus van Huissteden** [2], **Alexander Kholodov** [1,5], **Alexander Smirnov** [6], **Anton Kudryavtsev** [4], **Kanayim Teshebaeva** [2] and **Dmitrii Zamolodchikov** [7,8]

1 Institute of Physicochemical and Biological Issues in Soil Science of the Russian Academy of Sciences, Pushchino, Moscow Oblast 142290, Russia; elizaveta.rivkina@gmail.com (E.R.); tvishniv@utk.edu (T.V.); akholodov@mail.ru (A.K.)
2 Department of Earth Sciences, Vrije University of Amsterdam, 1081 HV Amsterdam, The Netherlands; j.van.huissteden@vu.nl (J.v.H.); k.teshebaeva@vu.nl (K.T.)
3 Department of Microbiology, University of Tennessee, Knoxville, TN 37996, USA
4 West-Siberian Filial, Trofimuk Institute of Petroleum Geology and Geophysics of Siberian Branch of Russian Academy of Sciences, Tyumen 625026, Russia; belonosov-a@mail.ru (A.B.); kudryavtsevae85@mail.ru (A.K.)
5 Geophysical Institute, University of Alaska Fairbanks, Fairbanks, AK 99775, USA
6 Industrial University of Tyumen, Tyumen 625000, Russia; smirnovas@tyuiu.ru
7 Faculty of Biology, Lomonosov Moscow State University, Moscow 119991, Russia; dzamolod@mail.ru
8 Center of Forest Ecology and Productivity of the Russian Academy of Sciences, Moscow 117234, Russia
* Correspondence: kraevg@gmail.com; Tel.: +33-(0)77-834-0399

**Abstract:** The gas shows in the permafrost zone represent a hazard for exploration, form the surface features, and are improperly estimated in the global methane budget. They contain methane of either surficial or deep-Earth origin accumulated earlier in the form of gas or gas hydrates in lithological traps in permafrost. From these traps, it rises through conduits, which have tectonic origin or are associated with permafrost degradation. We report methane fluxes from 20-m to 30-m deep boreholes, which are the artificial conduits for gas from permafrost in Siberia. The dynamics of degassing the traps was studied using static chambers, and compared to the concentration of methane in permafrost as analyzed by the headspace method and gas chromatography. More than 53 g of $CH_4$ could be released to the atmosphere at rates exceeding 9 g of $CH_4$ m$^{-2}$ s$^{-1}$ from a trap in epigenetic permafrost disconnected from traditional geological sources over a period from a few hours to several days. The amount of methane released from a borehole exceeded the amount of the gas that was enclosed in large volumes of permafrost within a diameter up to 5 meters around the borehole. Such gas shows could be by mistake assumed as permanent gas seeps, which leads to the overestimation of the role of permafrost in global warming.

**Keywords:** fluxes of $CH_4$; epigenetic cryogenesis; cryogenic transport; permeability of permafrost; methane accumulations; methane-hydrates; terrestrial seeps; pingo drilling

## 1. Introduction

The most recent estimation of the global methane budget includes the following listed sources of methane, in order of decreasing significance: the natural wetlands, geological sources (including oceans), freshwater sources (lakes and rivers), hydrates, and permafrost [1]. Permafrost, with its budget of 1 Tg $CH_4$ yr$^{-1}$, is the weakest source on this list today, and is considered to be more important in the near future. The thawing of permafrost provides a substrate and creates a favorable environment for the bacterial production of methane in wetlands and lakes [2]. It also provides an input of methane to

the carbon cycle from the degradation of gas hydrates, the release of gas from coal beds, traditional gas reservoirs, dispersed or locally accumulated methane in permafrost, and other geological sources [3–5]. Geological sources globally emit about 54 (33-75) Tg $CH_4$ $yr^{-1}$ [1], which is four times less than natural wetlands. However, being mostly point sources, they are usually stronger.

The geological sources in the permafrost zone usually deliver methane to the surface through conduits, ascending from deep horizons through the permeable strata related to faults, taliks, and deposits with a low degree of saturation by ice [6]. The conduits reaching the ground surface and continuously emitting methane are usually referred to as seeps. The capacity and variability of the seeps as assessed by observations both at the local and regional scale fully rely on different land classifications for making larger-scale extrapolations [3,4,7].

Terrestrial seeps in Alaska, which are mostly associated with lakes, create the flux of methane from deep sources as large as 0.2 - 4.5 g $CH_4$ $m^{-2}$ $yr^{-1}$ (3.7 g $CH_4$ $m^{-2}$ $yr^{-1}$ average) of lake surface, and has a strong variation between different sub-regions. About twice as much is emitted from thawing permafrost itself within lake or river-closed taliks [3]. A similar study conducted in northeastern Siberia revealed emissions of 33.7 g $CH_4$ $m^{-2}$ $yr^{-1}$ of lake surface and attributed them solely to microbial decomposition in thaw lakes [8]. The latest study from the East Siberian Sea shelf reports the surface emission of 49.1 g $CH_4$ $m^{-2}$ $yr^{-1}$ tracked from the sea bottom at a hotspot (area with high density of seeps) of 18,400 $km^2$ [9]. Earlier similar seeps were associated with the decomposition of sub-permafrost gas hydrates due to permafrost degradation on the shelf [4].

When the studies did not rely on the ebullition as a seep indicator, the findings of terrestrial seeps or gas shows were occasional, unless the methane seep resulted in the surficial forms as large as the Yamal crater. The latter is a gas blowout surface feature in continuous permafrost as evidenced by the elevated methane concentration in the air inside the Yamal crater [10], measured after an explosion. That gas accumulation happened through either cryogenic processes [5,11,12], the destabilization of the gas-hydrate layer [13], or ascending deep gases [14].

More often, the methane emits from boreholes causing the ebullition of drilling mud, drill kicks, or even fire [12,13,15]. Reviews of drilling reports documenting the gas shows provided a view on the broad geography of the phenomenon of gas shows from the topmost permafrost horizons in many regions of the Siberian Arctic. The gas shows' dynamics that have been studied in West Siberia provided data that they originated from methane accumulations at depths between 28–150 m. The emission of methane lasted from a few days to several months, gradually decreasing from the highest initial gas flux rate ranging from about 28 kg $CH_4$ $day^{-1}$ to 10,000 kg $CH_4$ $day^{-1}$ [12]. Another study reported methane concentrations reaching 77.5% vol. of borehole for several boreholes along the coast and in the shallow shelf of the East Siberian Sea [16].

A borehole acts as a conduit or a chimney, which is similar to natural disturbances or permeable sediments delivering gas from a source or lithological trap to the surface. Previous studies showed that epigenetic permafrost always contains methane in an average concentration of 2.7 mg $CH_4$ $m^{-3}$ (up to 47.4 g $CH_4$ $m^{-3}$) of frozen sediment as opposed to syngenetic permafrost, which in most cases did not have any detectable methane [17,18]. High concentrations of methane in permafrost deposits were found in redeposited and refrozen sediments of drained thaw lake basins in Central Yakutia [19], whereas moderate concentrations were detected in loams and the segregation ice of marine deposits on the Yamal Peninsula [20]. Methane in permafrost could be produced at temperatures below zero by methanogenic archaea [21], and it could be converted to interpore gas hydrate [22]. Methane in unfrozen sediments could move for several meters due to the freezing of the sediments via the mechanism of cryogenic transport and become accumulated locally in lithological traps [5,12]. Traps tend to be composed of coarse-grained sediments and have high active porosity [13].

A gas show should not be called a seep unless it has a signature of a deep source [7]. The most common signatures are the concentration of another gas (alkanes, $CO_2$, He, $H_2$, etc.) and the isotopic composition. If methane has $\delta^{13}C < -60‰$, it could be in most cases considered biogenic [23]. However, exceptions are not rare, especially with early mature natural gas [24]. The gas shows with the value of

$\delta^{13}C$ as low as –87‰, and values from –60‰ to –65‰ were reported for 10 gas-bearing fields in West Siberia at depths below 700 m [25]. The gas from shallower permafrost studied in Russia and Canada, which was both degassed from sediments [17,20] and emitted from accumulations [5,12,26], was mostly biogenic. Methane seeps in Alaska were reported to be thermogenic and of mixed origin [3], which was strongly distinct from the surficial methane sources in closed talik. Recently discovered crater-like methane seep in the Canadian High Arctic constantly emits methane of thermogenic origin [27]. However, the isotopic effects of source depletion and/or the process of oxidation could make the biogenic methane in the specific environment look isotopically thermogenic [23]. So, the isotopic composition alone is not an ultimate indicator, and the complex understanding of environmental conditions of methane formation, migration, accumulation, and degradation are always necessary for genetic interpretations.

Our review above shows that emissions from permafrost and sub-permafrost sources are poorly quantified compared to superficial methane fluxes. This might be the reason for their improper classification, because the physical forms and temporary locations of methane find themselves in the list of natural sources [1]. The gas-hydrates or permafrost could not be listed as separate sources in the methane budget, due to representing the cases of either geological or wetland sources, depending on the genesis of methane. Improper classification reduces the quality of extrapolations and forecasts. We hypothesized that there is a link between the gas shows, gas accumulations, methane hydrates, and dispersed methane in permafrost. In this paper, we attempt to find and explain these links based on our findings of the methane shows in epigenetic permafrost. We also analyze the dynamics of methane emission over time, and discuss the possible sources of methane accumulation and its concentration in the upper 30 m of permafrost on watersheds, lake depressions, and pingos on several sites in the Siberian Arctic.

## 2. Materials and Methods

Studies were carried out in two regions: one located in the oil and gas-bearing province in the Pur Lowland in West Siberia with a permafrost temperature of –2.5 °C; and another located in the Kolyma Lowland in northeastern Siberia, where the permafrost has temperatures from –6 to –10 °C (Figure 1). Studies in the West Siberia took place in the winter of 2017. Boreholes in the Kolyma Lowland were drilled during the summers (August–September) of 2007, 2013, 2015, and 2017. Quaternary to Pliocene frozen disperse sediments were disclosed by the rotary drilling down to a depth of 30 m. Rotary drilling was performed using the UKB-12/25, URB-2NT, and URB-4T drilling machines (ZIV, Vorovsky Machine-Building Plant, Russia), which were operated without any drilling fluids, with the outer diameters of the core-barrel ranging from 0.108 m to 0.127 m at the top to 0.055 m to 0.102 m at the bottom. Gradually smaller drilling diameters were chosen with the deepening of the boreholes to improve the performance in terms of the drilling rate and core deliverability. The depths of diameter changes were documented. Core extraction was either mechanical or pneumatic. The thermal and mechanical effects of drilling on permafrost spanned a zone that does not likely exceed 0.003 m around the borehole, assuming a similar thermal effect of drilling on the walls of the borehole and the core. Depending on the plasticity of the sediment, 0.001 m to 0.003 m of mangled soil was usually removed with a knife from the surface of the core to reach the frozen inner part of the core with an undisturbed structure. In the winter, when the outside temperature reached –17 °C, the core-barrel was subjected to warming with a blowtorch, which was thawed to about the same thickness of the outer layer of the core to provide easier extraction. The heat wave during the drilling and extraction of the core might change the thermal and pressure fields in the permafrost, which could partly recrystallize the pore ice, and thus induce fracturing or decompose gas hydrates. We assumed that this effect might not be propagated more than 0.028 m, which was half of the diameter of the thinnest core, because otherwise we would never find methane in these core samples.

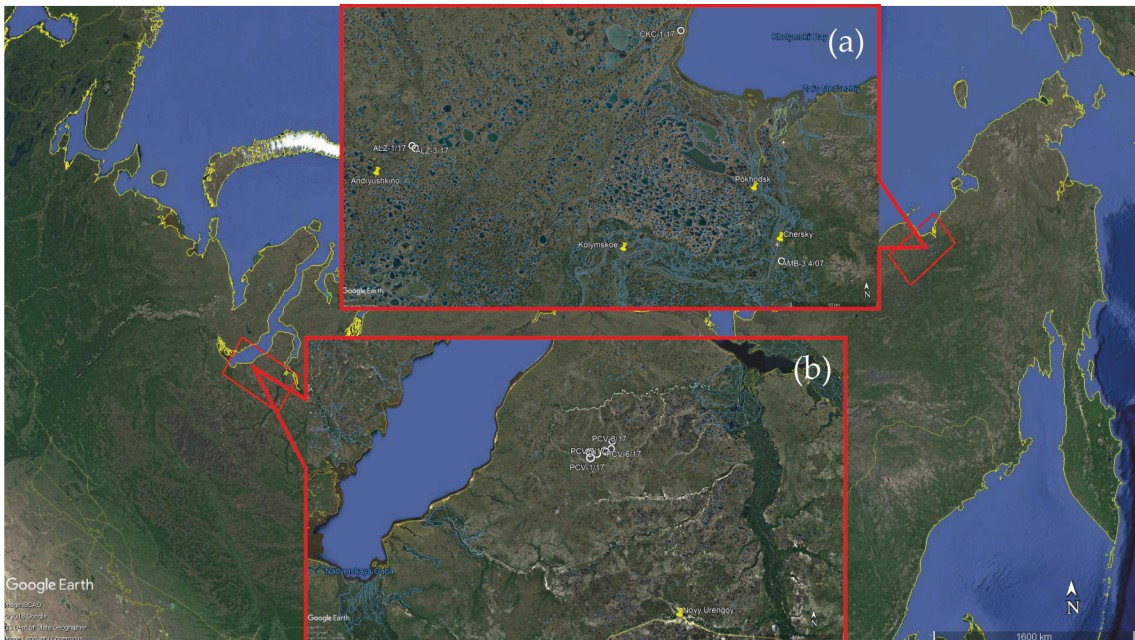

**Figure 1.** Location of study sites and boreholes in (**a**) the Kolyma Lowland, northeastern Siberia; (**b**) the Pur Lowland, West Siberia.

### 2.1. Methane Concentration in Permafrost

The $CH_4$ concentration, cryogenic structure, and basic physical properties of sediments were studied in the cores. The gas samples were collected in the field, weighed, and analyzed by the static headspace method [28]. Permafrost core samples of about 50 g were immersed in saturated NaCl solution in a 150-mL syringe. The volume of the permafrost sample was measured as a displaced volume of the solution with an accuracy of 1 $cm^3$. Then, the atmospheric air of the site was added and a sample was left to thaw, and was then crushed by intensive shaking. The gas mixture was then transferred through a needle to a rubber septa-sealed vial filled with saturated NaCl solution. At that point, brine was displaced by gas, with the excess of brine flowing out through a second needle.

The losses of methane during sampling were estimated using Equation (1) and the assumption that prior to degassing in saturated NaCl, the outer 0.001-m layer around the permafrost core sample thawed, and methane was released from this layer into the air through pores that were no longer cemented with ice. It was within $8.7 \pm 1.2$ % ($n$ = 189).

$$\Delta = (V_0 - V_1) \frac{\left(1 - \frac{\rho_0}{\rho_t(1+W)}\right)}{V_0},$$ (1)

where:

$\Delta$—losses from the outer 0.001-m layer of the sample;
$V_0$—sample volume, $m^3$, given the diameter of the syringe ($d_0$ = 0.038 m) and height of the sample ($H_0 = \frac{4V_0}{\pi d^2}$);
$V_1$—volume of frozen sample after 0.001 m is thawed (given the $d_1$ = ($d_0$ − 0.002) = 0.036 m; $H_1$ = $H_0$ − 0.002 m);
$\rho_0$—bulk density of permafrost;
$\rho_t$ = 2650 kg $m^{-3}$—normal particle density of quartz;
$W$—moisture content.

Losses of methane during headspace analysis were analyzed with multiple consequent headspace extractions from one sample, and using Equation (2) [28]. It was shown that $8.5 \pm 6.1$% (n = 11)

of the total methane amount is left in the soil solution after single extraction, with higher values corresponding to lower iciness.

$$\sum C = \frac{C_1^2}{C_1 - C_2},$$  (2)

where:

$C$—total concentration of methane in the sample;
$C_1$, $C_2$—concentration of methane after the first and second degassing, respectively.

The gas samples were kept in the vials until they were analyzed in the laboratory. The $CH_4$ concentration in the gas sample was measured with KhPM-4 (Khromatograph, Russia) and Crystall-2000 (Khromatek, Russia) gas chromatographs equipped with a flame ionization detector. Hydrogen and nitrogen were used as the carrier gases, respectively. The use of two different gas chromatographs allowed the estimation of the technical error variance, which was within 5%. Comparing the chromatographic signal from the sample to the signals of laboratory standards of 0.1% and 1% methane, the volumetric concentrations (units of volume) of methane in the headspace were found. Since the methane concentration in the atmosphere on the sampling sites did not exceed 0.1 ppm, which coincided with the precision of our chromatographic measurements, it was neglected, and all of the methane in the headspace was assumed to be extracted from the permafrost core sample. The volumetric concentration in the headspace was not equal to the concentration of methane in the permafrost soil, and the volume of the gas phase in the soil was uncertain. However, the amount of methane in the headspace was equal to the amount of methane in the permafrost soil, since it all came from the sample. The mass of methane in the permafrost sample was recalculated following the ideal gas law, as shown in Equation (3), and furthermore, it was related to the sample volume to obtain the mass concentration per unit volume of permafrost $C_s$ (g m$^{-3}$).

$$m = M \frac{1000 p V c}{RT},$$  (3)

where:

$m$—mass of methane in permafrost sample, g;
$M$ = 16.04 g mol$^{-1}$—molar mass of $CH_4$;
$p$ = 101325 Pa—normal atmospheric pressure (during sampling of gas);
$V$—volume of headspace, m$^3$;
$c$—measured concentration of $CH_4$, unit volume;
$R$ = 8.314 m$^3$ Pa K$^{-1}$ mol$^{-1}$—gas constant;
$T$—temperature of gas during sampling of headspace gas, K.

The permafrost sample volume was not measured when the CKC-1/17 borehole samples were degassed. The average densities for the stratigraphic units [29] of northeastern Siberia were used for the recalculation of the sample mass to the sample volume.

The overall losses and uncertainties related to chromatographic analyses were estimated to be 17.2 ± 8.0%. The concentration values reported here are not recalculated to reflect the losses of $CH_4$.

### 2.2. Emission of Methane

Dark polyvinyl chloride chambers with diameters of 0.115 m and heights of 0.25 m, and caps equipped with valves had been used to install on the borehole mouth. The gap between the outer diameter of a chamber and the diameter of a borehole was insulated with polyethylene. Boreholes were not absolutely concealed by chambers and could not catch all the gas due to the unconformities of the borehole mouth, although it might have prevented losing data and instruments due to the high pressure in the borehole created by the emitted gas. Due to the leakage on contact between the chamber

and the borehole, we expect an underestimation of the gas flux during the observation at a level not higher than the least detected flux density of 0.1 $\mu$g m$^{-2}$ s$^{-1}$. At the same time, the measured values of the initial methane concentration in the borehole and flux dynamics can be assumed to be reliable.

The valve of a chamber was connected to a Polaris-1001 (precision 1 ppm, Polaris, Russia), which is an optical detector of CH$_4$ equipped with a pump. The concentration of methane and temperature were measured in the air inside the borehole initially after chamber installation. The peak concentration that was shown by the instrument was accepted as the concentration of methane in the borehole. The maximum concentration was typically reached within three minutes. After a measurement, the valve was closed until the next measurement. The concentration of methane in the borehole, $C_b$, was calculated similar to $C_s$ using Equation (3), with $V = V_b$, the volume of the borehole was treated as the chamber volume. The average measured diameter of the borehole and borehole depth were used in the calculation of the volume.

Methane concentration inside the borehole secured with the chamber was measured occasionally, following the same procedure, with an irregular period from one day to one week after setting up a chamber. The total emissions of methane were measured until the concentration increment during follow-up measurements was below 10 ppm, except for PCV-4/17, where the measurements were interrupted due to technical reasons. The total mass of CH$_4$ emitted from the borehole was estimated by integrating the difference in concentration against time, following the trapezoidal rule in Equation (4):

$$E = V_b \sum_{k=1}^{N} \frac{\left(\frac{C_k}{t_k} + \frac{C_{k+1}}{t_{k+1}}\right)(t_{k+1} - t_k)}{2}, \tag{4}$$

where:

$E$—emission of CH$_4$, total mass of methane emitted from the borehole since chamber installation, g;

$C_k$—volumetric concentration of methane in the borehole measured at time $t_k$, g m$^{-3}$;

$t_k$—exposition time since chamber installation, s.

For the boreholes with a single measurement of methane concentration, when the gas was collected by a syringe through a 25-m long silicone tube deployed into the borehole, the emission of methane was assumed to be equal to the initial mass of the methane that was detected. Five samples were taken from the closed chamber to verify the concentrations measured by the methane detector in the field. Concentrations below 1000 ppm measured with a Polaris-1001 lay within the error interval of chromatographic measurements. For higher concentrations, it tended to underestimate the concentration measured with the chromatograph by an average of 50%. Accordingly, we corrected two values for methane measured in borehole PCV-7/17.

The methane flux density from borehole, $F$, was calculated by dividing the mass of the methane in it by the time of chamber isolation and the borehole mouth area.

To understand the links between the concentrations of methane in boreholes $C_b$, $F$, and $E$, and concentrations of methane $C_s$ found in sediments, the term "equivalent diameter of degassing" was introduced. All of the measured concentrations of methane in the core were averaged down to a depth at which the emission of methane was measured. The average concentration of CH$_4$ in sediments degassed from the core was used to find the volume of sediments containing the same amount of the gas as emitted from a drilling hole. We assumed that this volume of sediments forms the cylinder around the borehole, which must be degassed into it to make up the total volume of CH$_4$ emitted. We used the diameter of this cylinder, as shown in Equation (5), as an indicator of the volume of sediments, to better understand the power and longevity of the sources of CH$_4$ in permafrost. Thus, the equivalent diameter of degassing indicates a suggestive, not a real, volume of sediments that

contributed to the concentration of methane inside the borehole. The location of the $CH_4$ source in permafrost was assessed when the emissions from the borehole were measured at different depths.

$$D = 2\sqrt{\frac{C_b V_b}{C_s \pi H}},$$ (5)

where:

     *D*—equivalent diameter of degassing of permafrost around the borehole to meet the concentration of $CH_4$ in the borehole or the emitted amount of methane, m;

     $C_s$—concentration of methane in sediments averaged by borehole, mg m$^{-3}$;

     *H*—borehole depth, m.

### 2.3. Isotopic Studies

The same samples of gas were analyzed for $\delta^{13}C$ of methane using a GC Combustion Thermo Finnigan interface and Delta XL mass spectrometer (Thermo Fisher Scientific, Waltham, MA, USA).

## 3. Results and Discussion

### 3.1. Geological Sections

Boreholes in West Siberia were located at the boggy watershed plain, poorly drained by tributaries of the Khadutta River, which is the left inflow of the Pur River. Eight boreholes disclosed monotonous epicryogenic sands and loams of the Salekhard Formation (*amIIskh*) both on a remnant of an alluvial–marine plain about 90 m above sea level (a. s. l., boreholes PCV-2/17, PCV-2a/17, PCV-2b/17, PCV-2c/17, Figure 2a,c), and in widespread thermokarst depressions, approximately 20 m lower (the other PCV boreholes, Figure 2b,d–f, and Figure 3a–c). The latter were comprised of lacustrine and bog deposits, including icy sandy loam, loam, and peat in the topmost eight meters.

Two boreholes were drilled in the upper part of the slope of five to eight-meter pingos (PCV-6/17, PCV-7/17, Figure 3a,b), and one on palsa (PCV-8/17, Figure 3c). They disclosed loam and peat banded with transparent bubbly ice, which reached a thickness of 0.9 m. The watershed boreholes disclosed an infiltration talik composed by sand oversaturated with water at the interval of 1.4 m to 8.4 m (PCV-2/17, Figure 2a), and 0.6 m to 4.6 m (PCV-2a/17, Figure 2c). Confined aquifer was also disclosed under pingo (PCV-7/17, Figure 3b) at a depth of 28.2 m, and within three days filled the borehole up to 5.5 m.

Boreholes in East Siberia were located within the Kolyma Lowland on an alluvial–marine plain (CKC-1/17, 4 m a. s. l., Figure 3d), on a floodplain (ALZ-1/15, AMB-3,4/07, 4 m a. s. l., Figure 3e, [5]), and a remnant of a fluvial–lacustrine plain (ALZ-3/15, Figure 3f) to disclose the deposits of the Pliocene to late Pleistocene age by boreholes down to 20 to 25-m depth. The section consisted of epicryogenic alluvial sand of the Tumus Yar suite ($aN_2^{1-3}t$-*j*, ALZ-1/15, Figure 3e) and gravel of the Begunovka suite ($aN_2^{1-2}bg$, AMB-3,4/07 [5]). They were covered by the epicryogenic loam of the Olyor horizon (*lalE$_2$-I$_1$ol*, ALZ-3/15, Figure 3f) deposited within a floodplain, the Middle Pleistocene icy silt of the Keremesit horizon (*lalII-III$_1$krm*), and the deltaic sandy loam of the Kon'kovskaya suite (*amII$_2$kn*) on the Cape Chukochiy (CKC-1/17, Figure 3d). The syncryogenic late Pleistocene ice complex, the so-called Yedoma horizon (*lalIII$_{2-4}$yed*), was disclosed by ALZ-3/15, Figure 3f. Also, recent alluvial and marine sand and loam were sampled from ALZ-1/15, CKC-1/17, and AMB-3,4/07, as shown in Figure 3d,e, [5].

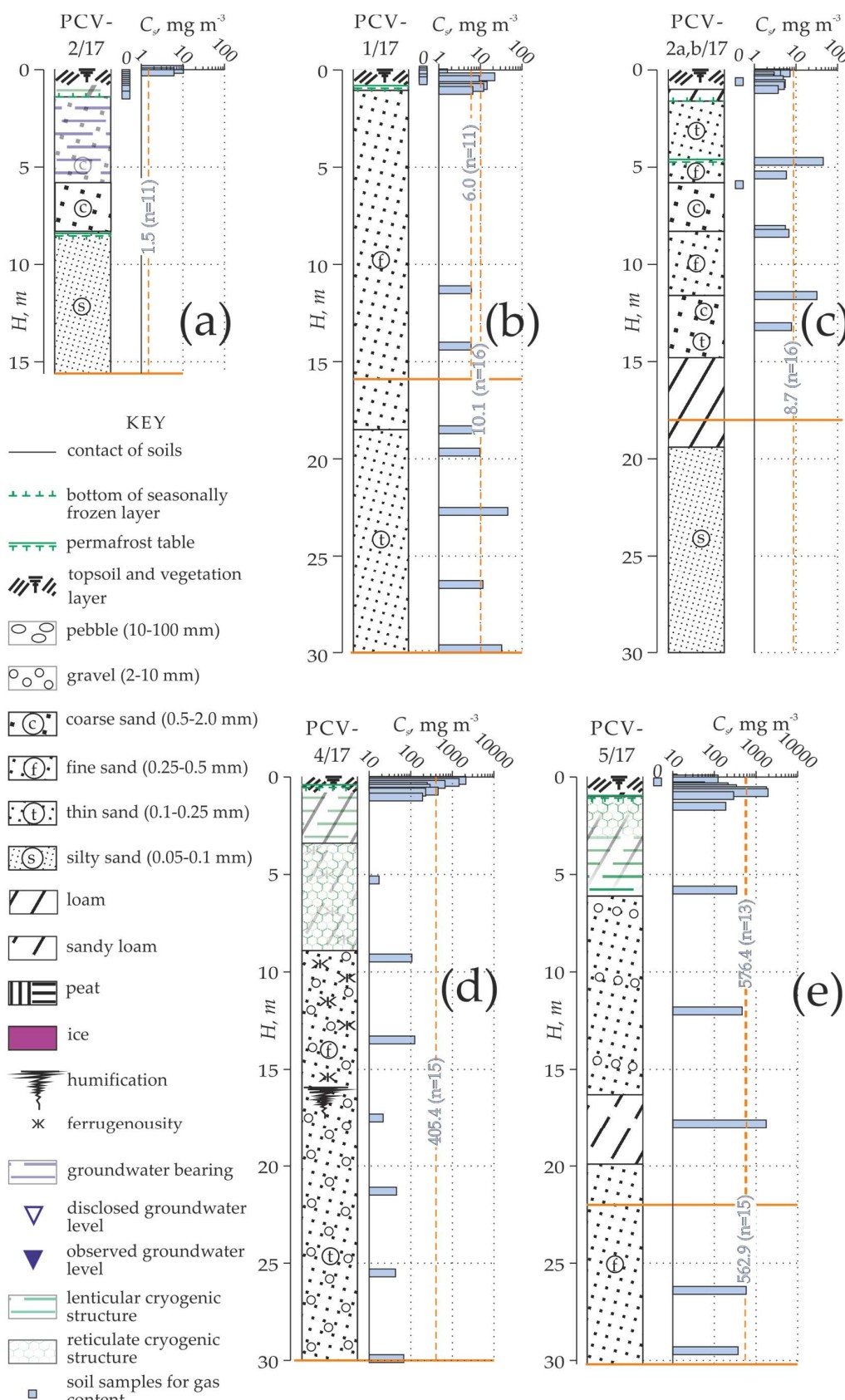

**Figure 2.** Geological sections and concentrations of methane in boreholes in the Pur Lowland (**a**) PCV-2/17, (**b**) PCV-1/17, (**c**) PCV-2a,b/17, (**d**) PCV-4/17, (**e**) PCV-5/17. Key for the geological sections.

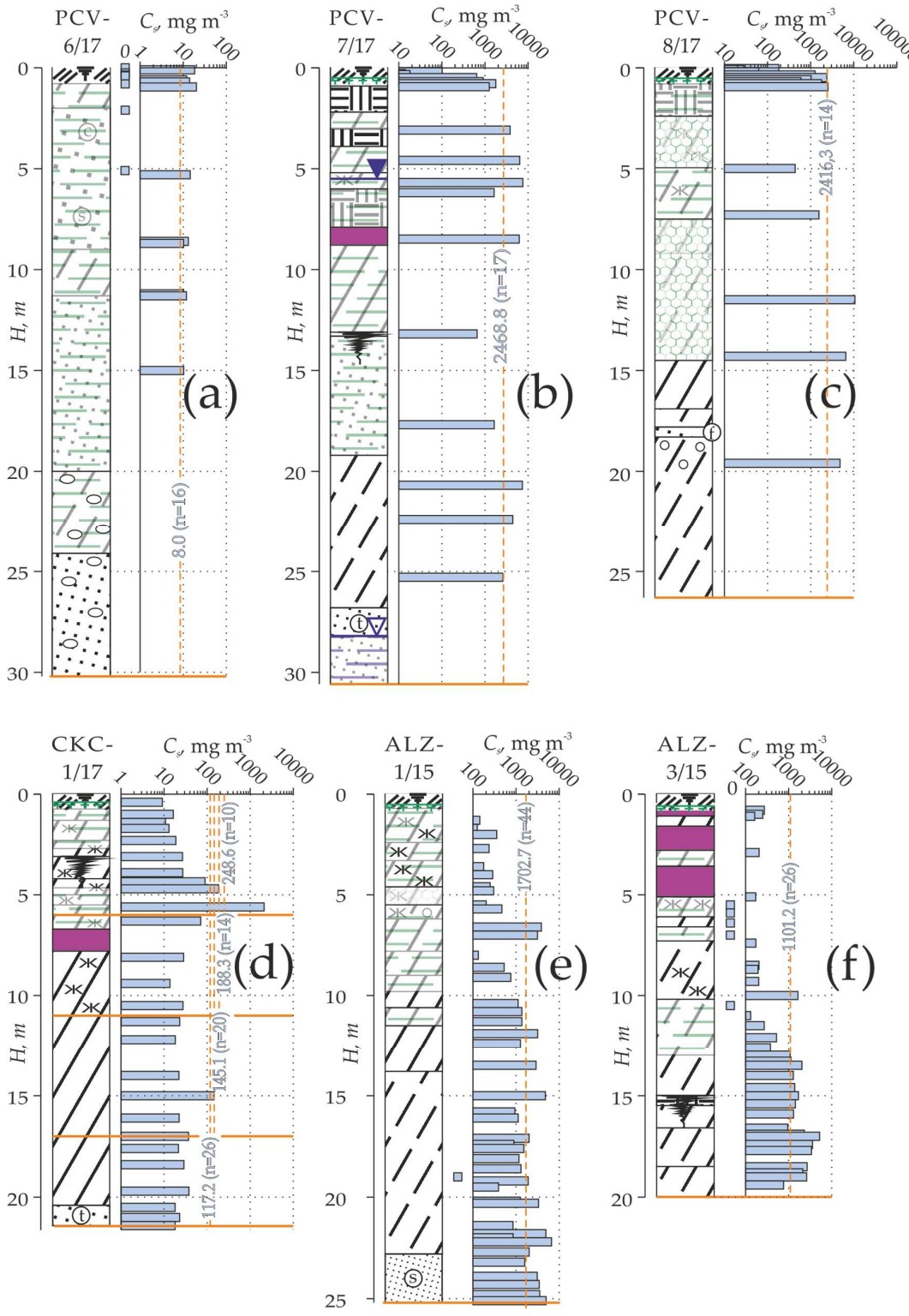

**Figure 3.** Geological sections and concentrations of methane in boreholes in the Pur and Kolyma lowlands (a) PCV-6/17, (b) PCV-7/17, (c) PCV-8/17, (d) CKC-1/17, (e) ALZ-1/15, and ALZ-3/15. Refer to Figure 2 for the key.

### 3.2. Methane Concentrations in the Cores

The concentration, $C_s$, of $CH_4$ in the sediments of West Siberia varied from 0 to 10.6 g m$^{-3}$, with an average of 0.7 g m$^{-3}$. The maximum value corresponded to the sample of icy lacustrine loam, which was collected from the core of PCV-8/17 on a palsa in the lake depression. The maximum

concentrations averaged along the core of 2.4 to 2.5 g $CH_4$ m$^{-3}$ were found in the boreholes PCV-7/17 and PCV-8/17 (Figure 3b,c).

Methane concentration in the cores of the Kolyma Lowland boreholes was exceptionally high. There was no methane detected only in the syncryogenic Yedoma horizon on the Alazeya River (borehole ALZ-3/15, Figure 3f). The highest concentration of methane in permafrost, about 20 g m$^{-3}$, was found in borehole AMB-3,4/07, which was drilled on the floodplain of the Kolyma Lowland in sediments of Oxbow Lake [5], with an average concentration of 6.2 g m$^{-3}$. Average values in all of the studied cores on Kolyma Lowland were above 1.5 g m$^{-3}$, as seen in Figure 3d,f. The only exception was the CKC-1/17 borehole, where the average $C_s$ was about 0.1 g m$^{-3}$, but the spike above 2 g m$^{-3}$ was found at 5.6-m depth, as shown in Figure 3d.

### 3.3. Initial Concentrations of Methane in Boreholes

As shown in Table 1, the concentration of $CH_4$ in boreholes, $C_b$, that was measured initially after drilling varied from 0 to about 600 g m$^{-3}$. The initial concentration of methane in the boreholes strongly depended on the average concentration of methane in the permafrost $C_s$ ($r = 0.83$, $n = 18$, $p < 0.05$), Figure 4. The concentration of methane in the boreholes was higher when the boreholes disclosed methane-rich sediments. The differences in $C_b$ occurred within the same borehole when sediments with higher (PCV-1/17, upper part of CKC-1/17) or lower concentrations of methane (PCV-5/17, lower part of CKC-1/17) were disclosed.

**Table 1.** Concentration of methane in boreholes $C_b$ and its relation to the concentration $C_s$ of methane in permafrost as measured in the core, and the equivalent diameter of degassing $D$ of permafrost around the borehole to meet the concentration of methane observed inside the hole. The diameters exceeding the weighted average diameter of the borehole $d_0$ are marked in bold.

| Borehole | H, m | $d_0$, m | $V_b$, m$^3$ | $C_b$, mg m$^{-3}$ | $C_s$, mg m$^{-3}$ | D, m |
|---|---|---|---|---|---|---|
| PCV-1/17 | 15.9 | 0.112 | 0.157 | 0.8 | 6.0 | 0.040 |
| | 30 | 0.119 | 0.333 | 2.6 | 10.1 | 0.059 |
| PCV-2/17 | 15.6 | 0.128 | 0.201 | 0.4 | 1.5 | 0.021 |
| PCV-2a/17 | 18 | 0.128 | 0.232 | 178.3 | 8.7 | **0.541** |
| | 23.4 | 0.124 | 0.284 | 4.4 | 8.7 | 0.083 |
| PCV-2b/17 | 30 | 0.122 | | 1.3 | | |
| PCV-2c/17 | 2 | 0.128 | | 1.3 | | |
| PCV-4/17 | 30 | 0.128 | 0.386 | 8.9 | 405.4 | 0.019 |
| PCV-5/17 | 22 | 0.125 | 0.270 | 3.1 | 576.4 | 0.009 |
| | 30.2 | 0.122 | 0.350 | 1.4 | 562.9 | 0.012 |
| PCV-6/17 | 30.2 | 0.113 | 0.300 | 0 | 8.0 | 0 |
| PCV-7/17 | 30.6 | 0.124 | 0.367 | 214.5 | 2468.8 | 0.043 |
| PCV-8/17 | 26.3 | 0.128 | 0.339 | 5.6 | 2416.3 | 0.006 |
| ALZ-1/15 | 25.2 | 0.069 | 0.093 | 409404.4 | 1702.7 | **1.064** |
| ALZ-3/15 | 20.0 | 0.072 | 0.081 | 79890.1 | 1101.2 | **0.610** |
| AMB-3,4/07 | 23.0 | 0.070 | 0.088 | 608285.9 | 6424.2 | **0.679** |
| CKC-1/17 | 6.0 | 0.090 | 0.038 | 115877.8 | 248.6 | **1.939** |
| | 11.0 | 0.082 | 0.058 | 301792.8 | 188.3 | **3.289** |
| | 17.0 | 0.074 | 0.074 | 54864.5 | 145.1 | **1.444** |
| | 21.4 | 0.071 | 0.083 | 57017.9 | 117.2 | **1.565** |

When borehole PCV-2a/17 disclosed (hydro)thermal talik below 1.3 m of frozen soil, the concentration of methane in the borehole, $C_b$, was an order of magnitude higher than in the permafrost (Table 1). The concentration of methane, $C_b$, was negligible in the borehole PCV-2/17, which was drilled in the same natural conditions nearby, but with talik saturated with water. On the contrary, the confined aquifer disclosed under pingo (PCV-7/17) had high initial concentrations of methane. When aquifers were disclosed, there was usually a fountain of water from the borehole with the drilling still ongoing,

and there was no possibility of measuring the emission of gases coming with the water. It is probable that the initial high concentrations of methane coming up with the water were lost.

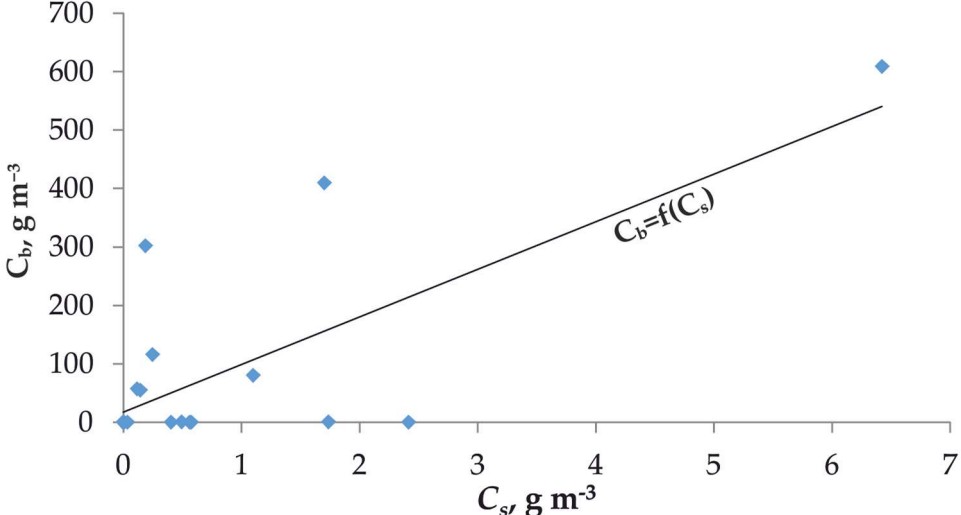

**Figure 4.** Scatter plot of the concentrations of methane found in the borehole initially after drilling $C_b$, and average concentration of methane in permafrost $C_s$.

Concentrations of $CH_4$, $C_b$, that were recorded in the boreholes of West Siberia were equivalent to the ones that could be formed due to the degassing of one to six cm of permafrost around the borehole axis, which is larger than topmost thawed layer of a core. However, when the borehole disclosed talik (PCV-2a/17, Figure 2c), the equivalent diameter of degassing $D$ fourfold exceeded the diameter of the borehole $d_0$.

Borehole PCV-2b/17 was drilled 0.5 m away from the borehole PCV-2a/17, and did not have such a large equivalent diameter of degassing. The borehole PCV-2c/17 was drilled 50 m away from PCV-2a/17 down to a depth of 2 m, for the purpose of getting through the upper layer of permafrost and collecting the methane from talik; it had a small concentration, $C_b$, of $CH_4$. All of the boreholes in West Siberia except PCV-2a,b/17 may have obtained their initial concentration, $C_b$, due to the effect of drilling on the walls of the boreholes. On the other hand, the excess amount of methane in the borehole could come from some part of the sediment column, which was not sampled, because the distance between the samples of permafrost for gas concentration measurements ranged from 0.05 m to 15 m. The sources of methane may be connected to the void of a borehole.

Boreholes in the Kolyma Lowland had the initial mass of methane in the borehole equal to the permafrost soil volume within $D = 0.6$ to 3.2 m around the borehole, exceeding the diameter of borehole $d_0$ by eight to nearly 45 times.

The equivalent diameter of degassing $D$ decreased with depth in borehole PCV-2a/17, increased in PCV-1/17 and PCV-5/17, and changed trend several times in borehole CKC-1/17. An equivalent diameter of degassing is an indicator for the localization of the depth intervals contributing to methane emissions. For example, in borehole CKC-1/17, we identified two horizons with increased methane emission within intervals where $D$ increases: an interval between six and 11 m gives maximum to initial concentration of methane, and the smaller source is located between 17 and 21 m (Figure 3d). Measuring the $CH_4$ concentration inside the borehole more often could help find methane-bearing intervals more precise. Studying these intervals thoroughly using extracted cores will enable us to better understand whether methane comes from gas hydrates or deposits with higher permeability.

*3.4. Dynamics of Methane Emission*

The dynamics of gas emissions were studied in the boreholes PCV-1/17, PCV-2/17, PCV-2a/17, PCV-4/17, and PCV-7/17 (Figure 5). Other boreholes were sampled only once. The long-term dynamics

of $CH_4$ fluxes were similar for all of the monitored boreholes. The initial concentration of methane in the boreholes and the density of methane flux were the highest right after drilling. The maximum density of methane flux from a borehole equal to 36 mg m$^{-2}$ s$^{-1}$ was registered within the first 20 min after the drilling of PCV-7/17 (Figure 5f). Fluxes from boreholes were gradually decreasing down to levels of 10 µg m$^{-2}$ s$^{-1}$ in two to five days, regardless of the initial maximum flux. As shown in Table 2, the average flux density exceeded 4 mg m$^{-2}$ s$^{-1}$. The average flux density, $F$, was linked with the initial concentrations of gas in the boreholes ($r^2 = 0.99$, $n = 6$, $p < 0.2$), as shown in Figure 6; thus, the initial concentration can be used for its assessment.

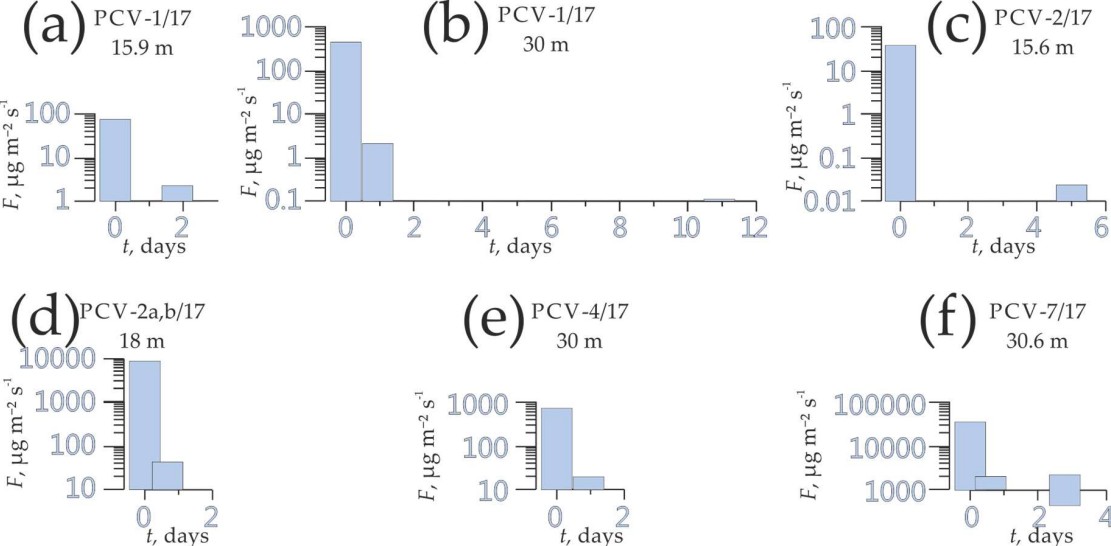

**Figure 5.** Changes in rates $v_b/t$ of methane discharged from boreholes during chamber measurements: (**a**) PCV-1/17 (15.3 m), (**b**) PCV-2/17 (30 m), (**c**) PCV-2/17 (15.6 m), (**d**) PCV-2a,b/17 (18 m), (**e**) PCV-4/17 (30 m), and (**f**) PCV-7/17 (30.6 m).

**Table 2.** Emission $E$ of methane from boreholes as measured in closed chambers, time of chamber exposition $t$, maximum $F_0$, and mean $F$ gas release rates and equivalent diameter of degassing of permafrost around boreholes $D$. Estimates of average fluxes calculated with the relationship on Figure 6 are italicized. The diameters excessing the drilling diameter are marked in bold.

| Borehole | | $t$, s | $F_0$, µg m$^{-2}$ s$^{-1}$ | $F$, µg m$^{-2}$ s$^{-1}$ | $E$, mg | $D$, m |
|---|---|---|---|---|---|---|
| PCV-1/17 | 15.6 | 155820 | 74.6 | 38.4 | 58.9 | **0.853** |
| | 30.0 | 864060 | 441.6 | 20.8 | 199.5 | **0.886** |
| PCV-2/17 | 15.6 | 427320 | 37.3 | 18.6 | 102.5 | **2.368** |
| PCV-2a/17 | 18 | 54540 | 8926.4 | 4484.8 | 3147.5 | **4.720** |
| PCV-4/17 | 30 | 82500 | 742.8 | 381.3 | 404.8 | **0.206** |
| PCV-7/17 | 30.6 | 177600 | 36587.3 | 3177.7 | 5801.0 | **0.403** |
| ALZ-1/15 | 25.2 | | | *6386.8* | | |
| ALZ-3/15 | 20 | | | *1246.4* | | |
| AMB-3,4/07 | 23 | | | *9489.3* | | |
| | 6 | | | *1807.8* | | |
| CKC-1/17 | 11 | | | *4708.1* | | |
| | 17 | | | *856.0* | | |
| | 21.4 | | | *889.6* | | |

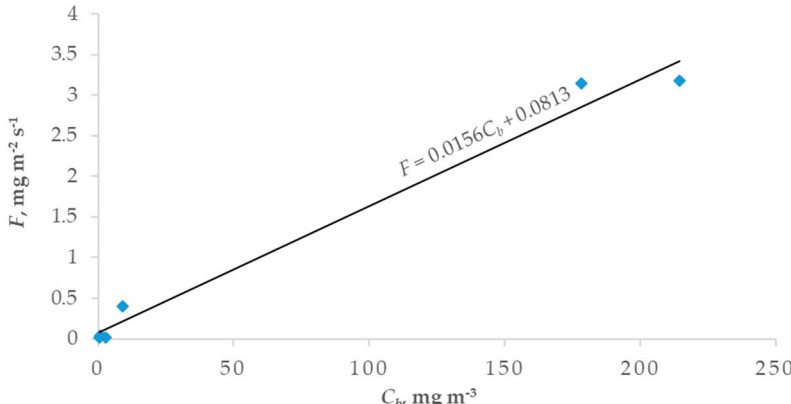

**Figure 6.** Scatter plot of the initial concentration $C_b$ of methane in borehole and the average flux from borehole *F*: linear relationship and its equation.

The measured and estimated average flux densities were compared to the studied sources of methane. Our findings were from 1.6 g m$^{-2}$ day$^{-1}$ to above 0.8·10$^6$ g m$^{-2}$ day$^{-1}$. If such a source of the flux is found and studied for less than 10 days, the researcher might come up with the flux estimate of 35.8 kg CH$_4$ m$^{-2}$ yr$^{-1}$ for the boreholes of West Siberia, and more than 0.1 × 10$^6$ kg CH$_4$ m$^{-2}$ yr$^{-1}$ for the Kolyma Lowland. The flux density of methane *F* emitted by the boreholes is up to three orders of magnitude higher than that reported for either Alaskan [3] or northeast Siberian thaw lakes [8] and sources on the shelf [9], and less but comparable with the gas shows reported for deeper boreholes on the Yamal Peninsula [13]. However, such extrapolations overestimate up to 10$^6$ times the measured emission, *E*.

The total mass of methane emitted from boreholes could only be found within the equivalent degassing diameter *D*, which exceeded the diameters of disturbance and the thawing of permafrost around the boreholes during drilling. The excess of *D* over the drilling diameter $d_0$ ranged within 2–40 times. For sediments with large $C_s$ such as those in PCV-4/17 and PCV-7/17, *D* was rather modest, while for the boreholes PCV-2/17 and PCV-2a/17, it reached several meters.

*3.5. Toward an Understanding of Methane Origin in Geologic Sections*

When D is much greater than the estimated size of a zone disturbed by drilling, there should be a source of emission that might be associated with either the methane accumulation in the form of a gas-enriched porous space, gas hydrate, or soils with high permeability conducting methane from some other unknown source. The cores were sampled and described in detail, and no visible inclusions looking or behaving similar to methane hydrates were found. However, the occurrence of the interpore diffused methane hydrate [22] cannot be excluded.

Strong links between concentration of methane in the core and initial concentration in the borehole support the hypothesis of the cryogenic mechanism of the methane accumulation [5,11,12]. On the other hand, the variability of the methane concentrations in permafrost can be caused by the gas migration in the changing permeability of sediments during and after freezing.

Not directly related to permafrost, but a good illustration of how the permeability works, could be the set of boreholes PCV-2a,b/17 and PCV-2c/17 disclosing the talik on watershed in the West Siberia. As *D* shows, the source of methane in the borehole PCV 2a/17 was likely the thawed material of the talik, which is a good location for the formation and accumulation of biogenic methane [5,30,31], with its anoxic conditions, above zero temperatures, and impermeable layer of frozen soil on the top. The methane storage in talik came into exchange with the atmosphere through the borehole thanks to the high permeability of the thawed sediments, where the pores' junctions were not constrained with ice. The lack of CH$_4$ in the nearby boreholes PCV-2b/17 and PCV-2c/17 might be caused by the discharge of a part or the whole storage of methane in talik through borehole PCV 2a/17. Imagine the behavior of the bubble in porous media under the bent bottom of a seasonally frozen layer. When it

was disclosed by the borehole, the bubble shrank, degassing into the borehole. However, a smaller portion of the bubble could be preserved in some positive structure of the bottom of the upper frozen layer. In the late summer, if it thaws, the gas might find its way to the atmosphere.

Similar to the thawed sediments, in permafrost, pores that are unfilled with ice work as the conduits connecting the borehole to some gas accumulation. In order to be able to behave that way, the gas accumulations should be under pressure. The pressure might be regulated by either freezing or water and gas flows. A highly pressurized system was disclosed in pingo by borehole PCV-7/17, where an aquifer disclosed at 28 m was set at 5.5 m below the surface, which was at the level of the pingo foot. This borehole had the gas show, which was the largest that we found in West Siberia. Having a low water solubility, methane might have migrated to the void space above the water. The low permeability of pingo could be suggested because of the multiple ice layers that were described in the core. This low permeability did not allow methane to penetrate in the layers of pingo, explaining the disproportion expressed in high *D* (Table 2).

The $\delta^{13}C$ was measured in four samples of methane emitted from boreholes, as shown in Table 3. The $\delta^{13}C$ values from three boreholes were below –80‰, which points to the biological origin of the methane. Methane from pingo PCV-7/17 was heavier. Nevertheless, this methane still fell in the zone of biogenic methane on the isotopic diagram, which is used as a standard nowadays, suggesting that it either represented a mixture of biogenic and thermogenic methane or an oxidized form of biogenic methane [23].

Table 3. Isotopic signature of methane in the boreholes.

| Borehole | $H$, m | $\delta^{13}C(CH_4)$, ‰ VPDB |
|---|---|---|
| PCV-2a/17 | 18 | −86.5 |
| PCV-7/17 | 30.6 | −64.4 |
| ALZ-1/15 | 25.2 | −84.9 |
| AMB-3,4/07 | 23.0 | −85.0 |

Large methane accumulations in the Kolyma Lowland were preserved due to the high ice content of the permafrost in the upper cover layers. Borehole AMB-3,4/07, as it was reported earlier [5] had 12 kyr younger methane compared to the age of sediments. The accumulation was likely formed by the cryogenic transport of methane. This methane was depleted in $^{14}C$, but according to $\delta^{13}C(CH_4)$, it had biological origin. In the other three cases, the location of methane accumulation was quite similar: methane accumulated in sand or gravel under an icy layer of silt. We could not call it a seep, because it has biogenic gas, which creates a lithological gas trap.

On the other hand, we cannot exclude the escape of early-mature gas [24] through conduits of permeable sand to pingo in West Siberia, especially given that pingos were reported to be often located above the geological gas conduits [14,32]. If it is really connected to a deeper source, then it is a seep, but we observed that the methane emission from it, although at higher initial rate, ended after two days. This might be caused not only by a lack of methane in the source, but also by a constrained conduit.

Methane from talik in PCV-2a,b/17 was not surprisingly biogenic, as it also should have formed in the soil and the talik itself, representing the surficial feature of (hydro)thermal origin.

Boreholes represent a very specific kind of permafrost disturbance, which has a very narrow conduit leading methane directly to the atmosphere, and is rare in natural types of disturbance. Natural conduits like faults usually are initially associated with depressions, which fill with water in the permafrost zone. Much of the gas passing through this environment must have dissolved, dissipated, become buried, or been microbially transformed, and thus did not reach the surface. Only young features related to gas emission such as boreholes, or gas explosion craters emit methane at the high rates that we observed. However, such types of emission are short-term. In our case, the emission did not exceed 10 days; for deeper boreholes, it may last for months. The Yamal crater

emitted methane no longer than several months, because no signs of ebullition were detected on its surface when it was examined [10]. Features in permafrost zone that emit large amounts of gas usually do not do so at a time scale of years.

Given the surface origin of methane in the gas shows, which occured both in gas-bearing West Siberia and non-gas-bearing Kolyma Lowland deposits, we should agree with the published research [3] that in cold continuous permafrost, the methane accumulations are preserved better than in areas that experienced glaciations. It is likely that in the latter regions, methane can emit through the permeable sediments to the air, contributing to the surficial fluxes. In both regions, we see that gas accumulations are mainly surface features, but in some cases, the geological origin is common. It was pointed out in our previous studies that methane accumulates in epigenetic permafrost, which was formed by the freezing of sediments that already passed the diagenetic stage [33] and were originally saturated with gas. The concentration of methane in such deposits is higher than in syngenetic deposits, which lack methane in most cases [17]. The epicryogenic freezing of methane-rich soils was considered to be the main mechanism of the gas dislocation and the formation of methane accumulations in permafrost [5]. The findings of excessive storages of methane that were reported here, which could be released from the boreholes, further supports the opinion that the history of permafrost formation should be taken into account in order to find the accumulations of methane in either form.

## 4. Conclusions

The gas shows from boreholes in epigenetic permafrost could be a strong source of methane, especially right after drilling. Based on the measured concentration in boreholes, we estimated flux through the borehole mouth from a 25-m deep borehole ranging from $1.6$ g $CH_4$ day$^{-1}$ m$^{-2}$ to $0.8 \times 10^6$ g $CH_4$ day$^{-1}$ m$^{-2}$. Such fluxes occur both in oil and gas-bearing regions and outside of them. The fluxes diminish to near-zero values within several days, so we do not find it appropriate to extrapolate the effect of any gas show to an annual timescale. Not all of them could be called seeps.

The emission of methane from boreholes could not be attributed only to the disturbing effect of drilling on permafrost. The volumes and dynamics of gas emission from boreholes point to the occurrence of traps filled with methane in permafrost. There are no visual features pointing to the existence of the trap in sediments. Both the surficial biogenic methane transported during freezing, and the diffusion/filtration of methane from deep-Earth sources via a system of conduits and traps represented by porous deposits unconstrained with ice can be associated with methane accumulation. The application of indicators such as the equivalent diameter of degassing provides an estimate of the size and, if monitored during drilling, might help to find the location of the source of methane in the borehole. The concentration of methane in permafrost samples also seems to be a good indicator of the occurrence of gas shows in boreholes.

$CH_4$ entrapped in permafrost could constitute some contributions to the fluxes of greenhouse gases in permeable permafrost. The disturbance of permafrost provides a path for the liberation of gas from the traps. The occurrence of lithological traps in permafrost is not ubiquitous, and could be studied using subsurface sounding, studies of permafrost evolution, gas amounts, and gas composition.

**Author Contributions:** Conceptualization, G.K. and E.R.; methodology, G.K., E.R., D.Z.; validation, G.K.; formal analysis, G.K.; investigation, G.K., E.R., T.V., A.B., A.K. (Anton Kudryavtsev), A.K. (Alexander Kholodov); resources, G.K., E.R., T.V., A.B., J.H., A.K. (Alexander Kholodov), D.Z.; data curation, G.K., E.R., T.V., A.B., J.H., A.K. (Anton Kudryavtsev); writing—original draft preparation, G.K.; writing—review and editing, G.K., E.R., T.V., J.H.; visualization, G.K., K.T.; supervision, G.K., E.R., T.V., A.S.; project administration, G.K., E.R., T.V., A.B., J.H., A.S., K.T.; funding acquisition, G.K., E.R., T.V., A.B., J.H., A.K. (Alexander Kholodov), A.S.

**Funding:** This research was funded by the Russian Academy of Sciences: grant numbers AAAA-A18-118013190182-3, AAAA-A18-118013190181-6, Russian Foundation for Basic Research 18-05-60279-Arktika, the National Science Foundation DEB-1442262, and the Nederlandse Organisatie voor Wetenschappelijk Onderzoek grant number ALW-GO/16-13.

**Acknowledgments:** The authors thank Alexey Nezhdanov for sharing ideas, taking part in the design of the study, and the organization of works. The authors are grateful to Thomas Röckmann, Carina van der Veen, and Elena Popa from Utrecht University and Tullis Onstott from Princeton University for carrying out laboratory

preparations and the measurements of stable isotopes. We also thank Oksana Zanina and Denis Shmelev for collecting field data on Chukochiy Cape and Alazeya River. We appreciate efforts to improve the text and representation of this publication, and kindly thank three anonymous reviewers.

**Conflicts of Interest:** The authors declare no conflict of interest.

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
