# Peer review of "Methane in Gas Shows from Boreholes in Epigenetic Permafrost of Siberian Arctic"

_geosciences, doi:10.3390/geosciences9020067_

Round 1

Reviewer 1 Report

The manuscript should be significantly improved before publication. The English should be extensively polished. All sections of the manuscript require restructuring and reworking. Methods section should be extended to cover the details for the headspace technique and chamber method for flux calculation from boreholes. Other comments are in the attached pdf file.

Author Response

We would like to thank the Reviewer for his efforts to critically evaluate our manuscript and for valuable commentaries in the text.

We did improved the article, added a lot to introuction and discussion. Added all relevant information to the Methods Section. Made more compact illustrations. Our response to reviewer may be find in the text of the edited manuscript, unless the edition was not that substantial, that the incorrect phrases were eliminated.

Reviewer 2 Report

This paper reports on observation results of methane gas from permafrost and natural gas hydrate within. The topic studying in this paper is timely. However, there are several minor points to be considered before publication. After the corrections considering following points, the paper would be worth publishing.

1)      Location map of the samples used is useful for better understanding of potential readers.

2)      More detailed captions and instructions on Figs.1-6 are necessary.      

3)      It is necessary to add reference numbers within the text of manuscript.

Author Response

We would like to thank the Reviewer for his optimistic view on our manuscript. Nevertheless we substantially reworked the text, graphics of the paper, and improved the technical issue with disappeared references prepared with EndNote. 

Reviewer 3 Report

The manuscript provides only experimental measurement. No extensive analysis is done based on the measurement. The manuscript looks like experimental report, not a scientific paper.

Author Response

We thank the Reviewer for the criticism he expressed to our paper. Your comments were valuable to us, and we tried to do our best to make the paper look like a study. Text, illustrations and tables were substantially reworked to make reading clear, informative, and perhaps interesting.